# TED: A Pretrained Unsupervised Summarization Model with Theme Modeling and Denoising

## ABSTRACT

Text summarization aims to extract essential information from a piece of text and transform it into a concise version. Existing unsupervised abstractive summarization models use recurrent neural networks framework and ignore abundant unlabeled corpora resources. In order to address these issues, we propose TED, a transformer-based unsupervised summarization system with pretraining on large-scale data. We first leverage the lead bias in news articles to pretrain the model on large-scale corpora. Then, we finetune TED on target domains through theme modeling and a denoising autoencoder to enhance the quality of summaries. Notably, TED outperforms all unsupervised abstractive baselines on NYT, CNN/DM and English Gigaword datasets with various document styles. Further analysis shows that the summaries generated by TED are abstractive and containing even higher proportions of novel tokens than those from supervised models.

## 1 INTRODUCTION

Summarization refers to the task of condensing a document into a shorter version. Summarization models can be categorized into two classes: abstractive and extractive. Extractive models select sentences from the input article as the summary. Such process ensures a basic level of grammaticality and accuracy, but also limit the model to copying. In contrast, abstractive models summarize documents using tokens and phrases that may not be found in the input article, a process requiring an advanced ability to refine, paraphrase and re-organize information (See et al., 2017; Narayan et al., 2018).

Like most machine learning algorithms, summarization models can also be divided into supervised and unsupervised categories. Supervised approaches require in-domain parallel data, i.e. both input articles and corresponding reference summaries must be present for training (Hermann et al., 2015; Liu & Lapata, 2019). Unfortunately, high-quality paired data are not always available across different text domains and styles. Moreover, considering the fact that summarization is not an easy task even for people, reliable human-labeled data are also difficult to obtain. Therefore, several unsupervised summarization approaches have been proposed, which do not require reference summaries for the target domain. We introduce these methods as follows.

**Unsupervised extractive models.** TextRank (Mihalcea & Tarau, 2004) encodes sentences in the article as nodes in an undirected graph. The weights of edges are measured by sentences similarity. The centrality of a node (sentence) is computed by PageRank (Brin & Page, 1998) to decide whether a sentence should be included in the final summary. Zheng & Lapata (2019) advances upon TextRank by using BERT (Devlin et al., 2018) to compute sentence similarity and build graphs with directed edges decided by the relative positions of sentences.

**Unsupervised abstractive models.** Baziotis et al. (2019) leverages differentiable sampling and optimizes by re-constructing the input article from the generated summary. Chu & Liu (2018) proposes a similar idea in the multi-document summarization setting. Wang & Lee (2018) uses adversarial training and reinforcement learning to make the summary human-readable. Févry & Phang (2018) adopts denoising autoencoders originally used in sentence compression. However, most of these models are only tested on datasets with considerably small article/summary length. Also, previous models usually utilize the recurrent neural networks (RNNs). However, transformers (Vaswani

et al., 2017; Devlin et al., 2018) have shown superior performances over RNNs on various NLP tasks, including machine translation, reading comprehension, sentiment analysis, etc.

In this paper, we present TED, an unsupervised abstractive summarization model with theme modeling and denoising that uses a transformer-based encoder-decoder structure and the pretraining leverages large scale unlabeled corpora. Our main contributions are two-fold as follows.

First, we leverage the lead bias in news articles for model pretraining. The lead bias is introduced by the journalistic convention of writing using an inverted pyramid structure, placing the most important information in the beginning of an article. We propose to use the leading sentences as the target summary and train the model to predict it during pretraining. In this way, we can utilize large-scale unlabeled corpora. Without any finetuing, the model pretrained in this way on 21.4M news articles can yield better performance than most existing unsupervised methods.

Second, to finetune on specific datasets, TED is further trained with a theme modeling loss and a denoising autoencoder. The role of the theme modeling module is to make the generated summary semantically close to the article. The module uses a semantic classifier trained using a discriminative objective function. Furthermore, to optimize on the generated summary tokens, we adopt the Gumbel-Softmax (Jang et al., 2016) estimator to replace the non-differentiable $\arg\max$. The denoising autoencoder has been previously used in unsupervised machine translation (Lample et al., 2017) and sentence compression (Févry & Phang, 2018), and we employ it to help the model extract salient information from corrupted text.

Also, instead of classical word tokenization, we adopt the SentencePiece tokenization (Kudo & Richardson, 2018) to alleviates the long-standing out-of-vocabulary (OOV) problem in language generation tasks (Luong et al., 2014; Sennrich et al., 2015).

We test TED on several benchmark datasets. The experimental results show that TED outperform all unsupervised abstractive baselines on all datasets. For example, on the CNN/DM dataset, it outperforms the state-of-the-art unsupervised abstractive model by more than 9 ROUGE-1 points and compares favorably with most unsupervised extractive models. We further show that TED is capable of generating novel words and phrases in the summaries, and is a highly abstractive system even compared with supervised systems.

## 2 METHODOLOGY

In this section, we will go through the model structure of TED, i.e. the transformer encoder and decoder, theme modelling and the denoising autoencoder. The overall architecture of TED is illustrated in Fig. 1.

### 2.1 TRANSFORMER ENCODER AND DECODER

Previous unsupervised summarization methods are based on the sequence to sequence (seq2seq) model (Sutskever et al., 2014) that primarily uses the RNN model. As the transformer structure (Vaswani et al., 2017) has been successfully used in a large number of NLP tasks, our model employs the multi-layer transformer encoder-decoder architecture. We follow the standard transformer design in our network and refer readers to Vaswani et al. (2017) for more technical details. Denote the number of layers (i.e., Transformer blocks) as $L$, the number of self-attention heads as $H$ and the hidden size as $N$. We explore two different configurations in experiments, 4 layers 4 heads (4L4H) with $N = 512$ and 10 layers 8 heads (10L8H) with $N = 720$.

Denote the input article token sequence as $X = \{x_1, x_2, ..., x_n\}$, and each token is transferred to a vector by a trainable embeddings matrix $\boldsymbol{V}$. The output from transformer encoder $E$ is a sequence of encoded vectors $E(X) = \{\boldsymbol{u}_1^E, \boldsymbol{u}_2^E, ..., \boldsymbol{u}_n^E\}$. The decoder can be viewed as a conditional language model to generate the summary. Given $k$ input summary tokens $W = \{w_1, w_2, ..., w_k\}$, the cross attention layer in the decoder $D$ attends with encoder outputs $\{\boldsymbol{u}_i^E\}_{i=1}^n$. The decoder outputs are $D(\{w_1, w_2, ..., w_k\}) = \{\boldsymbol{u}_1^D, \boldsymbol{u}_2^D, ..., \boldsymbol{u}_k^D\}$. The probability distribution over the vocabulary for $w_{k+1}$ is given by:

$$P(w_{k+1}|w_{1:k}, x_{1:n}) = \text{softmax}(\boldsymbol{V}\boldsymbol{u}_k^D) \tag{1}$$

In our model, the text are not tokenized by spaces but by the SentencePiece (Kudo & Richardson, 2018) model, in order to address the challenging out-of-vocabulary (OOV) words issue. Efforts have been made to address this issue at the cost of losing semantic information, such as mapping OOV words to a special "UNK" token. To mitigate the open vocabulary problem, we adopt SentencePiece, a data-driven method that trains tokenization models from sentences in large-scale corpora. The advantage of the SentencePiece model is that its subwords can cover all possible word forms and the subword vocabulary size is controllable. In our experiments, we train a SentencePiece subword vocabulary of size 32,000.

Note for supervised summarization models, the inputs to the decoder are the groundtruths/reference summary tokens; for unsupervised learning, input tokens are generated in the previous pass. More details are available in section 2.3.1.

## 2.2 PRETRAINING WITH UNLABELED CORPORA

Leveraging large scale unlabeled text corpora to pretrain models has been proven as an effective method in multiple NLP tasks (Devlin et al., 2018). However, such approach has not yet been utilized in text summarization.

News articles follow an inverted pyramid structure, i.e. front loading the most salient information. This so-called "lead bias" for news summarization is so strong that See et al. (2017) have shown that using the first 3 sentences in a news article as a summary can score higher than many sophisticated deep learning models. Although this poses a great challenge to previous research, we leverage this property in our favor in the pretraining phase.

For a news article, we set the target summary to be the first three sentences. This allows the model to exploit the structural bias of the news domain and infer the most important information using the background materials in the remainder of the article. For pretraining, we obtain three years of online news articles from 2016 to 2019 via an industrial search engine. The search engine indexes major online news domain, for instance, New York Times and Bloomberg. Then we collect the parsed articles within the 2016-2019 time range as the raw data. Note that this time span does not overlap any of three test datasets we use in this paper, therefore the pretraining should not lead to data leakage in test.

Next we conduct data cleaning to remove irrelevant distracting content and filter out articles whose top three sentences do not form a good summary. First, many news articles begin with media names, reporter names, dates or other irrelevant information for summarization, e.g. "New York (CNN) –", "Adam Smith, June 3rd 2018:". We automatically clean these using regular expressions. Second, we only include articles whose top three sentences contain between 10 and 150 words, and remaining sentences contain between 150 and 1,200 words. Third, we try to remove articles for which the first three sentences may not contain the major information in the article. We use a simple and easy-to-compute metric: overlapping words. We compute the portion of non-stopping words in the top three sentences that also appear in the rest of an article. A higher ratio indicates that the rest of the article is likely to elaborate on the beginning part. We keep those articles with this ratio of overlapping words higher than 0.65.

Finally, we end up with 21.4M articles. We randomly sample 12,000 from the data for validation. We conduct pretraining for 10 epochs and pick the model with the best ROUGE-L score on the validation set. The pretraining idea is also used in (Anonymous, 2020). After pretraining, we finetune TED on target datasets in an unsupervised manner. This includes two modules: *theme modeling* and *denoising autoencoder*.

## 2.3 THEME MODELING

Theme modeling aims to make the generated summary semantically close to the input article. We employ differential sampling to enable optimization on generated summaries and train a classifier to improve the semantic relatedness between the summary and article.

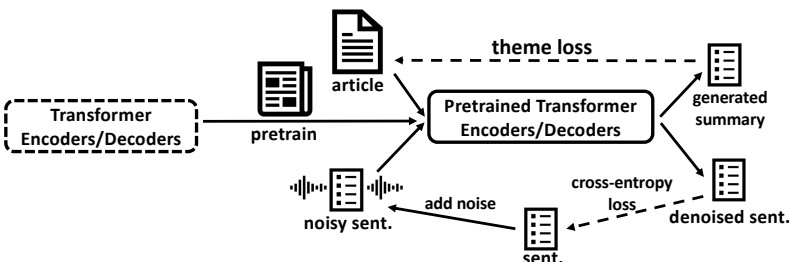

Figure 1: Overall structure of our model. TED first pretrains on news articles and then finetunes with theme modeling and denoising. (from left to right).

### 2.3.1 Differentiable Sampling

In order to optimize the network on output summaries, we need to make the generation of summary tokens differentiable. Recall the probability distribution of token $w_{k+1}$ is $P(w_{k+1}|w_{1:k}, x_{1:n}) = $ softmax$(\boldsymbol{V}\boldsymbol{u}_k^D)$. Let $\boldsymbol{\pi}$ denote $P(w_{k+1}|w_{1:k}, x_{1:n})$. One can use $\arg\max$ on $\boldsymbol{\pi}$ to obtain the token $w_{k+1}$ in the forward pass, however, it is not differentiable in the gradient back-propagation. Although one can get around by obtaining the embedding of $w_{k+1}$ as a weighted sum of the vocabulary embeddings $\boldsymbol{V}$, this results in an undesirable gap between the forward pass in training (weighted sum) and inference (discrete sampling). To solve this issue, we employ the straight-through Gumbel-Softmax estimator (Jang et al., 2016) as in Yang et al. (2018); Baziotis et al. (2019). Specifically, the forward pass in training still uses $\arg\max$ sampling, but for gradient computation, the following Gumbel-Softmax distribution is used as a differentiable approximation for the $\arg\max$ operation:

$$\tilde{\boldsymbol{\pi}_i} = \frac{\exp(\log(\boldsymbol{\pi}_i) + g_i)/\tau)}{\sum_{j=1}^{k} \exp(\log(\boldsymbol{\pi}_j) + g_j)/\tau)} \tag{2}$$

where $g_1, \cdots, g_k$ are i.i.d samples drawn from the Gumbel distribution $G(0, 1)$ and $\tau$ denotes the softmax temperature. As shown in Jang et al. (2016), as $\tau \to 0$, the Gumbel-Softmax distribution converges to the categorical (one-hot) distribution; as $\tau \to \inf$, the Gumbel-Softmax distribution converges to the uniform distribution. Although this gradient estimator is biased, we find that this method works well in practice. We choose $\tau = 0.1$ based on the CNN/DM validation set and use this value in all the experiments. Denote the input article as $\boldsymbol{d}$, the generated summary as $\boldsymbol{s} = \{w_1, w_2, ..., w_m\}$. The generation of $\boldsymbol{s}$ follows the recursive process that input $w_{1:k}$ to the transformer decoder to obtain $w_{k+1}$, then input $w_{1:k+1}$ to compute $w_{k+2}$ and so on. The first input token $w_1$ is always the special beginning token [START].

### 2.3.2 Encoder Transformer as a Semantic Classifier

We frame the semantic similarity problem in a discriminative setting. As the generated summary may be off the article theme at the beginning of training, we add sentence pairs from the article to facilitate similarity computation.

Concretely, during training, we pick two consecutive sequences of tokens $\boldsymbol{a}_1$ and $\boldsymbol{a}_2$ from an article to form a positive sequence pair $\{\boldsymbol{a}_1, \boldsymbol{a}_2\}$. Second, sequence $\boldsymbol{b}_1$ is chosen from a random article from the dataset to form the negative sequence pair $\{\boldsymbol{a}_1, \boldsymbol{b}_1\}$. Following Devlin et al. (2018), each sequence pair is packed into one single sequence by inserting a special token [SEP] between them and adding trainable segment embeddings. A special classification token [CLS] is also added to the beginning. As shown in Fig. 2, the packed sequence is fed as input into TED's encoder. The output vector associated with the token [CLS], is then classified into similar/distinct categories by a two-layer fully connected network. We use the following cross-entropy loss to optimize the encoder. Note that the theme modeling loss does not involve the transformer decoder.

$$\mathcal{L}_{theme} = -\log(p(y = 1|\boldsymbol{a}_1, \boldsymbol{a}_2)) - \log(p(y = 1|\boldsymbol{s}, \boldsymbol{d})) - \log(p(y = 0|\boldsymbol{a}_1, \boldsymbol{b}_1)) \tag{3}$$

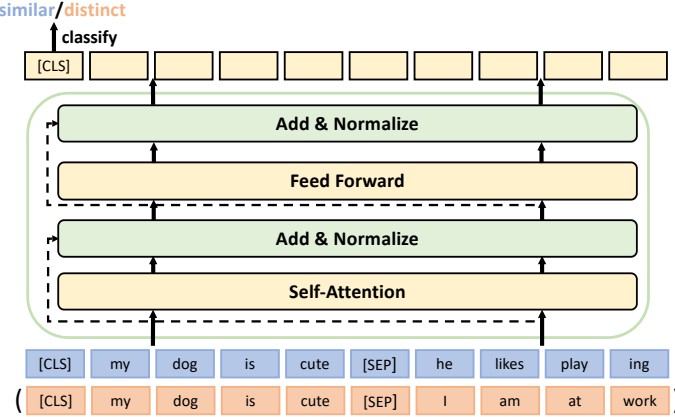

Figure 2: Theme modeling is essentially a semantic classifier. The input sentence pair is first processed by adding a "class" token in the beginning and a "separation" token in between. Then the sentence pair is fed into the transformer encoder, and then a linear classifier.

### 2.4 DENOISING AUTOENCODER

The idea of denoising autoencoder (Vincent et al., 2008) has been used in unsupervised machine translation (Artetxe et al., 2017; Lample et al., 2017) to prevent the model learning to merely copy every input word one by one. This denoising process imitates text simplification and helps to refine essential semantic information.

In detail, a sequence of $n$ consecutive tokens $\boldsymbol{x}$ from the input article is injected with two types of noise. First, we insert noisy tokens sampled from other articles in the same dataset into the original sequence at random positions, obtaining a new sequence with length $n'$, where $n'$ is 40%-50% larger than $n$. Next, similar to Lample et al. (2017), the sequence is slightly shuffled by applying a permutation $\sigma$ such that $\forall i \in [1, 2, \cdots, n'], |\sigma(i) - i| \leq k$, where the permutation distance $k$ is set to be 20% of the length of $\boldsymbol{x}$. The final corrupted sequence is denoted as $\boldsymbol{x}'$.

The TED model is trained to recover the original token sequence given the corrupted sequence:

$$\mathcal{L}_{denoise} = CE(\boldsymbol{x}, \text{TED}(\boldsymbol{x}'))  \tag{4}$$

where $CE$ denotes the mean of token-level cross-entropy loss. $\text{TED}(\boldsymbol{x}')$ denotes the sequence of probability distribution outputs $\{\boldsymbol{\pi}\}$ from the decoder with inputing $\boldsymbol{x}'$ to the encoder and $\boldsymbol{x}$ to the decoder for teacher forcing.

The final objective function is the mean of Eq. (3) and Eq. (4):

$$\mathcal{L}_{\text{TED}} = \frac{\mathcal{L}_{theme} + \mathcal{L}_{denoise}}{2}  \tag{5}$$

## 3 EXPERIMENTS

### 3.1 IMPLEMENTATION DETAILS

For pretraining, we use a dropout rate of 0.3 for all inputs to transformer layers. We use RAdam (Liu et al., 2019) as the optimizer, with a learning rate of $10^{-4}$. Also, due to the different numerical scales of the positional embedding and initialized sentence piece embeddings, we divide the positional embedding by 100 before feeding it into the transformer.

For unsupervised finetuning on specific datasets, the learning rate is set to $2 \times 10^{-4}$ and dropout ratio stays the same as in pretraining. The batch size is 16, and the vocabulary embeddings are also updated in the training process. During test, we generate the summarization from trained encoder and decoder by beam search.

| Dataset | # docs | avg. document | | avg. summary | |
|---|---|---|---|---|---|
| | | words | sen. | words | sen. |
| CNN/DM | 11,490 | 641.9 | 28.0 | 54.6 | 3.9 |
| NYT | 4,375 | 1,290.5 | 50.7 | 79.8 | 3.5 |
| English Gigaword | 1,937 | 29 | 1 | 8 | 1 |

Table 1: Average document and summary length in number of words and sentences on NYT, CNN/DM, and English Gigaword datasets (test set).

## 3.2 Results

We evaluate our model on three summarization datasets: NYT, CNN/DM and English Gigaword. The text statistics on these datasets are summarized in Table 1. Numbers of NYT and CNN/DM are collected from Zheng & Lapata (2019).

We compare TED with the following baselines: Brief (Wang & Lee, 2018), SEQ[3] (Baziotis et al., 2019), GPT-2 (Radford et al., 2019), TextRank (Mihalcea & Tarau, 2004), PACSUM (Zheng & Lapata, 2019), PGNet (See et al., 2017) and REFRESH (Narayan et al., 2018). These models cover both unsupervised and supervised categories, and include abstractive and extractive methods.

We measure the quality of generated summaries by ROUGE F1 score (Lin, 2004), including unigram (ROUGE-1), bigram (ROUGE-2) and longest common subsequence (ROUGE-L).

The **NYT** dataset (Durrett et al., 2016) contains 110,540 news articles with 100,834/9,706 train/test split. Following Liu & Lapata (2019), we choose 4,000 examples as the validation set and filter out examples with summaries of fewer than 50 words. As demonstrated by the results in table 2, the unsupervised fine-tuning of TED improves upon the pretrained model by 2.75%/1.06%/2.37% on ROUGE-1/ROUGE-2/ROUGE-L respectively. Note that ROUGE metric prefers extractive systems that preserve original phrasing (See et al., 2017). Considering this factor, TED achieves results that are competitive with unsupervised extractive baselines and surpasses all unsupervised abstractive models.

The **CNN/DM** dataset (Hermann et al., 2015) is composed of articles from CNN and Daily Mail, and uses associated highlights as reference summaries. We use the same training, validation and test split (287,227/13,368/11,490) as other baselines. Similar to See et al. (2017) and Liu & Lapata (2019), input articles are truncated to 500 tokens. Results are shown in Table 2. TED with a larger model size (10L8H) outperforms all unsupervised abstractive methods and compares favorably with unsupervised extractive baselines. Note that TED outperforms GTP-2, a powerful transformer-based language generation model pretrained on large scale webpage textual data, by siginificant margins. Again, TED further improves upon pretrained models on both 10L8H and 4L4H configurations.

For the **English Gigaword** sentence compression dataset, the input text is the first sentence of a news article, and the reference summary is the article's headline. The size of train/val/test is 3.8M/189k/1,937 respectively, after filtering out data examples with articles containing only "UNK" tokens. As shown in Table 3. TED outperforms all the unsupervised baselines.

## 4 DISCUSSION

### 4.1 ABLATION STUDY

The ablation studies shown in Table 4 verify the effectiveness of each module in TED. Training the transformer encoder-decoder from scratch yields reasonable performance. Pretraining on large-scale data results in more than 10% improvement on all three metrics on training TED from scratch. Pretraining plus either theme modeling or denoising improves upon the pretrained model by more than 2%. The full TED model, pretraining with theme modeling and denoising, produces the best result overall.

| Model | CNN/DM | | | NYT | | |
|---|---|---|---|---|---|---|
| | R1 | R2 | RL | R1 | R2 | RL |
| *Unsupervised Abstractive* | | | | | | |
| TED 10L8H (ours) | **38.73** | **16.84** | **35.40** | **37.78** | **17.63** | **34.33** |
| Pretrained 10L8H (ours) | 38.38 | 16.49 | 35.08 | 35.03 | 16.57 | 31.96 |
| TED 4L4H (ours) | 34.38 | 9.56 | 30.10 | - | - | - |
| Pretrained 4L4H (ours) | 31.20 | 10.05 | 27.80 | - | - | - |
| SEQ[3] | 23.24 | 7.10 | 22.15 | 17.85 | 3.94 | 19.53 |
| Brief | 28.11 | 9.97 | 25.41 | - | - | - |
| GPT-2 | 29.34 | 8.27 | 26.58 | - | - | - |
| *Unsupervised Extractive* | | | | | | |
| LEAD-3 | 40.50 | 17.70 | 36.70 | 35.50 | 17.20 | 32.00 |
| TextRank + tf-idf | 33.20 | 11.80 | 29.60 | 33.20 | 13.10 | 29.00 |
| TextRank + skip-thought | 31.40 | 10.20 | 28.20 | 30.10 | 9.60 | 26.10 |
| TextRank + BERT | 30.80 | 9.60 | 27.40 | 29.70 | 9.00 | 25.30 |
| PACSUM + tf-idf | 39.20 | 16.30 | 35.30 | 40.40 | 20.60 | 36.40 |
| PACSUM + skip-thought | 38.60 | 16.10 | 34.90 | 38.30 | 18.80 | 34.50 |
| PACSUM + BERT | **40.70** | **17.80** | **36.90** | **41.40** | **21.70** | **37.50** |
| *Supervised Abstractive & Extractive* | | | | | | |
| SUMO | 41.00 | **18.40** | 37.20 | 42.30 | **22.70** | **38.60** |
| PGNet | 39.50 | 17.30 | 36.40 | **42.70** | 22.10 | 38.00 |
| REFRESH | **41.30** | **18.40** | **37.50** | 41.30 | 22.00 | 37.80 |

Table 2: ROUGE $F_1$ scores on NYT and CNN/DM datasets. R1/R2/RL stands for ROUGE-1/ROUGE-2/ROUGE-L respectively. Best results in each unsupervised category is in bold. Results of other models are obtained from original papers or running open-sourced software.

| Model | R1 | R2 | RL |
|---|---|---|---|
| TED 10L8H (ours) | **25.58** | **8.94** | **22.83** |
| Pretrained 10L8H (ours) | 25.23 | 8.84 | 22.56 |
| TED 4L4H (ours) | 24.59 | 8.10 | 21.91 |
| Pretrained 4L4H (ours) | 22.52 | 7.46 | 20.09 |
| LEAD-8 | 21.86 | 7.66 | 20.45 |
| SEQ[3] | 25.39 | 8.21 | 22.68 |
| Brief | 21.26 | 5.60 | 18.89 |

Table 3: Results on the English Gigaword dataset. Numbers are collected from original papers. The best performance is in bold.

| Model | R1 | R2 | RL |
|---|---|---|---|
| train from scratch | 24.49 | 4.41 | 20.14 |
| pretrained only | 35.03 | 16.57 | 31.96 |
| pretrained w/ theme modeling | 37.16 | 18.18 | 34.15 |
| pretrained w/ denoise loss | 37.48 | 17.83 | 34.05 |
| full model | 37.78 | 17.63 | 34.33 |

Table 4: Ablation study of different components in TED on the NYT dataset. We test with the 10L8H model configuration.

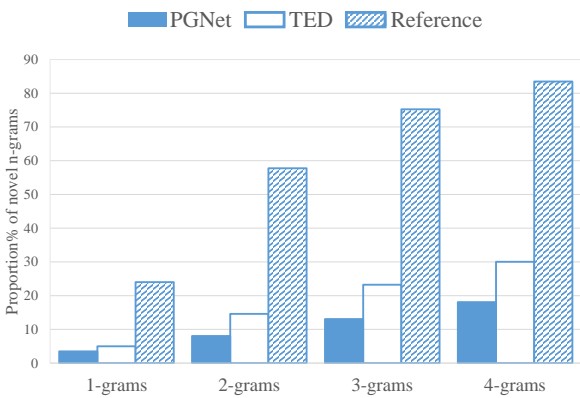

Figure 3: Proportion of novel grams in summaries on the CNN/DM test set. We compare three systems, TED, PGNet and reference summaries. Numbers of PGNet are computed from its publicly released output.

## 4.2 MODEL ANALYSIS

**Novel N-grams.** To examine how abstractive TED is, we compute the proportion of novel N-grams in the summary output (Fig. 3). The reference summary and the output from PGNet are included for comparison. Although TED is unsupervised, it includes more novel grams than the supervised model PGNet. The reference summaries have the highest proportion of n-grams.

**Example.** We showcase a sample summary from CNN/DM dataset along with the input article and the reference summary (Fig. 4). As shown, TED is able to capture and organize the essential information into fluent language. We attribute the grammatical correctness to the pretraining process and the denoising autoencoder. However, we also note that although TED manages to recognize the temporal information related to reported event (a few hours after Fox news reports), it makes a mistake by summarizing as "a few hours after a report about roberts' research was released. . . ". It shows that fact cross-checking is a potential future research direction.

---

**Article**
after exposing potential security risks with airlines' in-flight entertainment systems, one of the top experts on counter-threat intelligence in the world was pulled off a flight by fbi agents. chris roberts, who featured in a string of fox news reports, was yanked off his plane after it landed in syracuse, new york, on wednesday night by two fbi agents and two uniformed officers. roberts, who works for security intelligence company one world labs, was questioned for the next four hours ...

**TED Summary**
chris roberts, who works for security intelligence company one world labs, was pulled off a plane in syracuse, new york, on wednesday night by two fbi agents and two uniformed officers.
the incident occurred only a few hours after a report about roberts' research was released by the government accountability office earlier this week.

**Reference**
chris roberts of one world labs grabbed after plane landed in syracuse. two fbi agents spent four hours questioning him about cyberhacking. agents confiscated electronic devices and computer files from roberts. he flew in to give talk at aerospace conference about plane vulnerabilities. roberts featured on fox news' on the record with greta van susteren. regarded as one of the world's top experts on counter-threat intelligence."

---

Figure 4: An example of a generated summary by TED. The reference summary and parts of the input article are also included.

## 5 CONCLUSION

In this paper, we propose TED, an unsupervised abstractive text summarization model. First, we introduce an effective and powerful pretraining approach leveraging the lead bias in news articles. We then develop a finetuning scheme to induce the semantic similarity between summaries and input articles, together with a denoising autoencoder. Experiments across three datasets show that TED outperforms unsupervised abstractive baselines. For future work, we would like to encode the criteria of relevance, informativeness and importance proposed in Peyrard (2019) into TED. Fact cross checking is another interesting direction as mentioned in the section 4.2.

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
