# OpenReview forum: "TED: A Pretrained Unsupervised Summarization Model with Theme Modeling and Denoising"
_ICLR.cc/2020/Conference — Reject_

### Official Review · AnonReviewer2 · 2019-10-23
**Official Blind Review #2**

**Rating:** 8

**Review:**

POS-DISCUSSION
I thank the authors for their answer. I updated my score assuming ryxAY34YwB does not exist, and would encourage authors to discuss in more details the relationship with MeanSum if this gets accepted

PRE-DISCUSSION

This is an important contribution for the field of unsupervised summarization. "Unsupervised *" is trendy in NLP so this is a timely contribution. Furthermore, doing this for summarization is important because of the cost of getting gold summaries and the model used in translation is harder (impossible?) to adapt to this setting where there is information loss in one direction.

However, I find major drawbacks in the current state of this paper. They are best related to the three contributions the author claim:
 - Contribution3: the use of BPE. "BPE for X", with X being an NLP task can hardly count as a contribution today. If we are counting who did it first, then this is taken at least by Liu & Lapata 2019 through their use of BERT
 - Contribution1: leveraging the lead bias for pre-training. This is a great idea! However, this seems to be covered by an accompanying paper (ICLR submission ryxAY34YwB) which is not referenced. Because of common paragraphs and experimental setting I am assuming there is an overlap of the author sets in two papers. PLEASE CORRECT IF THIS IS NOT THE CASE. As you don't get to claim the same contribution twice, this contribution should go all to the benefit of the other paper.
 - Contribution2: the use of combining reconstruction loss and theme loss for summarization is another great idea. However, the paper that introduced this for summarization (as far as I know) is not cited nor compared too (MeanSum: https://arxiv.org/abs/1810.05739). This seems like a major issue considering the similarity in the approach (including the use of the straight-through Gumbel softmax estimator).

Other comments:

 - Being a growing topic of study, I appreciated in particular the care taken to report a number of other approaches. Could you please clarify which version of ROUGE was used in each case? There are significant differences in the different implementations being used.
 - Please also specify the version of ROUGE you used.
 - Your numbers in Table 2 do not coincide with Table 3 of ryxAY34YwB (eg: LEAD-3 for CNN/DM). Can you explain?
 - Your ablation study (Sect 4.1) focuses on CNN/DM (NOTE: the caption of Table 4 says NYT, but the number correspond to CNN/DM. I guess this is an error), where the topic & reconstruction loss indeed helps. However this is not the case for NYT, where LEAD-3 actually beats any of your approach. This is not mention nor discussed.
 - The example of Fig 4 reveals a major problem. The summary states an incorrect fact: the gov accountability had indeed released a report earlier that week; but this was NOT a few hours before the reported incident. What happened a few hours before was a report on Fox News.


In a summary: a good idea combining ideas of ryxAY34YwB and adapting MeanSum. However, this is in my opinion not enough material for a full paper.

**Experience Assessment:**

I have published one or two papers in this area.

**Review Assessment: Checking Correctness Of Derivations And Theory:**

N/A

**Review Assessment: Checking Correctness Of Experiments:**

I carefully checked the experiments.

**Review Assessment: Thoroughness In Paper Reading:**

I read the paper thoroughly.

---

> ### Author Response · Authors · 2019-11-12
> **To reviewer #2**
>
> We appreciate your comments! Please find our response below.
>
> About the concern on our contributions.
> (1) About SentencePiece. We have removed the claim from the major contributions. To the best of our knowledge, TED is one of the initial attempts to use SentencePiece in unsupervised text summarization.
>
> (2) Thanks for pointing that out. We have added the reference to that paper. We believe the usage of denoising and theme modeling in summarization are still innovative as discussed more in the next response.
>
> (3) Thanks for mentioning MeanSum and we have referenced it in the paper. The reasons why we didn’t include MeanSum are:
>
> First, MeanSum is for multi-document summarization, while the baseline models we are comparing are for single document.
>
> Second, the denoising in TED is quite different from the reconstruction idea in MeanSum. In TED’s denoising, the corrupted text are input to the transformer and the model is trained to filter the added noises. Note the original (clean) text is not used as inputs or seen by TED in the forward pass. However, the reconstruction process in MeanSum follows that it inputs multiple documents to RNN, generate the summaries (the encoded reviews), and then reconstruct each document from the summaries.
>
> Third, the same reconstruction idea is also used in a baseline single document summarization model SEQ3 (NAACL 19) that we compared with in the paper, which is published at almost the same time as MeanSum (ICML 19). Similar to MeanSum, SEQ3 tries to reconstruct the the single document from the generated summary. As shown in table 2, TED outperforms SEQ3 by significant margins.
>
> TED is innovative compared with both MeanSum and SEQ3. First MeanSum and SEQ3 both use RNN, while TED builds on transformer. Second, although both MeanSum and SEQ3 have a loss to make make the summary similar to the input article, it is implemented as the classical cosine similarity. In contrast, TED innovatively encodes the similarity by the transformer encoder in a BERT-style.
>
> About other comments:
> (1) Most of the performances of baseline models are directly taken from the original paper. After searching their paper, open-sourced code (if available) and by personal communications, we found that PacSum, TextRank (from the PacSum paper), SEQ3, Brief, GPT-2, SUMO, REFRESH, PGNet use ROUGE-1.5.5.
>
> (2) The ROUGE version we use is ROUGE-1.5.5, same as mentioned above.
>
> (3) We have corrected the numbers in Table 2 in the revised paper. Please refer to the newest table for the performance.
>
> (4) The ablation study in table 4 is on NYT dataset. The full TED model, pretrain w/ theme modeling and pretrain w/ denoise all outperform the lead-3 baseline now.
>
> (5) Fact/common sense checking would be an interesting future direction. Our model manages to recognize that there are time-related information it still needs improvement on delivering factual information. We’ve added the analysis to section 4.2.

---

### Official Review · AnonReviewer3 · 2019-10-23
**Official Blind Review #3**

**Rating:** 8

**Review:**

Paper's Claims

The paper introduces a new unsupervised abstractive summarization approach called TED, using a Transformer encoder and decoder. Their main contributions are as follows:
1) Pretraining the encoder and decoder on news articles using the first beginning as the target summary.
2) Fine-tune on other datasets using so-called theme modeling, and separately a denoising loss.
3) TED's performance is claimed to significantly improve over GPT-2 while not being too far from the best unsupervised extractive summarization results.

Decision

Edit: After revisions and discussions, I recommend we accept this paper.

I am leaning towards accepting this paper mostly because of the contribution #1 above. Unsupervised learning using large quantities of text that have the property of being typically written in a style that synthesizes information in the first 1-3 sentences is a powerful idea. That the performance is improved compared to other unsupervised abstractive summarization confirms the importance of this approach.

However the importance of and justification for the fine-tuning steps are comparatively much more limited in my opinion. Also, some important details about the preprocessing for pre-training appear to be missing and they could be quite important.

Detailed arguments for decision

I view this effort as aiming to reproduce the BERT approach in the context of abstractive summarization, which is a good idea. The most clever contribution is in leveraging un-labeled text using the first few sentences as the target summary for pretraining. The results of just this part are already beating previous approaches, while not requiring any in-domain data, which is quite powerful.

However, some relatively important details regarding the methodology are omitted or only glossed over and it would greatly contribute to making this work more reproducible if the details were included (see my detailed notes below, notably regarding section 2.2).

On the fine-tuning steps, I have several worries. First, why not fine-tune using supervised learning, as would be the analog to the BERT approach? Instead the authors go out of their way to do in-domain unsupervised learning, which provides a boost, yes, but still doesn't compare positively to extractive and/or supervised methods. Second, why not perform the theme modeling and denoising also -- or rather only -- on the unlabelled pretraining data? Why should it be done on the in-domain fine-tuning data instead (while not using the most valuable piece of in-domain information, namely the example summaries)? After all, it's a fully unsupervised approach and it can actually be performed on any text at all, whether a summary for it exists or not.

Again regarding the unsupervised approach, and to push the BERT analogy further, I'm wondering why not initialize the pretraining model with a BERT-style trained model? After all we could imagine building a system that adds more and more in-domain characteristics sequentially: first pretrain a BERT model, then fine-tune to summarization using what this paper calls pretraining, and then finally fine-tune again to a specific summarization domain.

So, to conclude, I find that this paper goes in the right direction and introduces important ideas for pretraining and fine tuning unsupervised abstractive summarization models, but that some decisions about how to use the various ideas (theme and denoising but no supervised learning, in-domain vs during pretraining) have not been explored enough.

Extra notes

page 2, second line: pretrainleverages (typo)
section 2.1: fix first sentence to make it an actual sentence.
section 2.2: "we obtain three years of online new articles ... via a search engine" please be more specific about your methodology.
section 2.2: You should double check more throughly that there is no data leakage in test. There could be articles about the same exact events, years apart, for example. I doubt that this would be a big effect, but there are easily ways to find highly similar articles between the pretraining data and test data to make sure.
section 2.2: "Next we conduct following data cleaning" fix (typo?). Also that sentence probably belongs to the next paragraph.
section 2.2: Why did you pick the values that you did for the preprocessing heuristics (such as between 10-150 words, 150-1200 words, 3 sentences and not 2 or 1 or 4, the ratio 0.65, etc.)? Were other values tried?
section 2.2: You mention you end up with 21.4M articles. How many were there to start with? What's the filtering ratio?
section 2.2: You mention that you pick the model with the best ROUGE-L score on the validation set. How many models were there? What was different between them?
section 2.2, OOV Problem: the information in this whole subsection would fit better in 2.1 where 'tokens' are left generic without specifying which type of token you're considering.
Figure 1: I find the upper part of this figure very confusing. Why are there arrows going from the encoder/decoder to a summary, to theme loss, to article and back to encoder/decoder? It's important that the summary is never seen by the theme loss otherwise it's not unsupervised anymore, and I also don't see why the arrow would go through article *after* theme loss. I assume there must have been a mistake, please fix.
section 2.4: "the sequence is slightly shuffled by applying a permutation /sigma such that ..." The formula given here tells me that all token indices are shuffled with another token within a window k. That seems like a lot of moving around, and also depending on the implementation a token from the beginning could possibly end up at the very tail of the sentence by being picked iteratively again and again, thus falling outside the permutation distance k. Please provide more details on how this is done and a justification for why it was decided to do it this way.
Section 3.1: I'd like to know how long (preferably number of words, or at least number of wordpiece tokens) the summaries generated are. What determines how long they are, is it a fixed size, or the model decides to stop on his own (or when hitting some limit), or something else?
section 4.2: Do you have any idea why your unsupervised approach yields more novel n-grams than a the supervised model you compare against? This can be good as much as it can be bad, in that it could be going off-track. Yes humans have high novelty, but high novelty in itself isn't necessarily good. I don't find the argument that have more novel ngrams is intrinsically, necessarily good, compelling. If I'm wrong, then it would be nice to have better explanation in the paper.




**Experience Assessment:**

I have published in this field for several years.

**Review Assessment: Checking Correctness Of Derivations And Theory:**

I assessed the sensibility of the derivations and theory.

**Review Assessment: Checking Correctness Of Experiments:**

I assessed the sensibility of the experiments.

**Review Assessment: Thoroughness In Paper Reading:**

I read the paper thoroughly.

---

> ### Author Response · Authors · 2019-11-12
> **To reviewer #3 (part 1)**
>
> To reviewer3:
> We appreciate your detailed and helpful comments. Due to the character limit, we reply with two separated posts (part 1 and 2).
>
> About the fine tuning steps.
>
> (1) The reason why we did not finetune with ground-truths is we want to design an unsupervised model. The motivation is in practice one has very limited or even no ground-truths summarizations for a dataset. Also high-quality summarizations are harder to obtain compared with other labelling data, e.g. semantic class. It is ecause summarization requires the advanced ability such as paraphrasing and information extraction, which indeed are not simple for humans. Therefore, we propose an unsupervised approach that does not rely on labelled summaries.
>
> (2) In the theme loss, TED generates summaries recurrently, i.e. using the previously generated tokens to predict the next one. This is because ground-truths summaries are not available and teacher-forcing cannot be used. This process is time-consuming in the pretraining with 21.4M examples, but is feasible for in-domain fine-tuning where data are limited. Also the unsupervised fine tuning techniques (theme modeling and denoising) is to address the scenario that ground-truths are unavailable. However, in the pretraining, the ground-truths are available (lead-3 sentences) so we train the transformer encoder and decoder with classical teacher forcing.
>
> (3) About BERT initializations. We train the transformers from scratch because, first, we have enough amount of training data to do so, i.e. 21.4M article-summary pairs. Second, BERT is not specifically designed for summarization tasks, while our pretraining is. The experiment results (table 2, pretrained) also shows that our pretraining is powerful and competitive.

---

> ### Author Response · Authors · 2019-11-12
> **To reviewer #3 (part 2)**
>
> Regarding the extra notes (with the same order as in “Extra notes”):
>
> (1) (2) Sorry about the typos. We have fixed them.
> (3) The search engine indexes major online news domain, for instance, New York Times and Bloomberg. Then we collect the parsed articles within the 2016-2019 time range as the raw data.
> (4)  We understand your concern about data leakage. We went through the three test sets and did not find significantly overlapped articles as in the pretraining.
> (5) Thanks for pointing it out. We have revised it.
> (6) Some explanations for the heuristic values selections:
>
> 150 and 1,200 words: Articles with very long content are filtered them mainly to reduce memory consumption. Short articles are filtered since the information might be too condensed and not suitable for summarization pretraining.
>
> 10 and 150 words: Some leading sentences are extremely short, e.g. one or two words phrases.Those are filtered since they have too little information to be reasonable summaries. Longer leading sentences are removed to reduce the pretraining time.
>
> 0.65: The overlap ratio is an indicator of the amount of information that the leading sentences maintain. For instance,  in CNN/DM dataset, the median of the overlapping ratio of non-stopping words between golden summary and the article is 0.87, and the ratio between the first 3 sentences and the rest of the article is 0.77 (median). Setting the number at 0.65 makes the final training set size fit with the available computation resources and ensures that the leading sentences contain enough information.
>
> We mean to have demanding filtering criteria since we want high-quality pretraining data. We didn’t try other settings since pretraining is a time-consuming process.
>
> (7) We start with about 407 million articles. The filtering ratio is about 95%. We’ve also added this information to the paper.
>
> (8) We train one model for 10 epochs. After each epoch, the model is evaluated on validation data. We pick the check points with the highest ROUGE L.
>
> (9) About OOV. It is a good idea. We have edited and moved the paragraph to section 2.1
>
> (10)  About Figure 1. Sorry about the confusion. The “summary” refers to the generated summary from the transformer encoders/decoders, not the groundtruths summaries. The process follows that the article is input to the transformer encoder/decoders and a summary is generated. Then we compute the theme loss using the generated summary and the article. We’ve changed the text label “summary” in figure to “generated summary” to avoid the confusion.
>
> (11) About sequence shuffling. Here is how we generate the permutations (the variable perm) of the indices using numpy. Assume the length of the sequence is L, and the window size is k.
> ids = np.arange(L)
> noise =  np.random.uniform(0, k, size = L)
> tmp = ids + noise
> perm = tmp.argsort()
> For tokens in the beginning, e.g. the first token, since there are at most k -1 elements smaller than tmp[0] in tmp, so the first token is at most shuffled to the kth position.
>
> The motivation of shuffling is as follows. The information is to extract and summarize is scattered across an article. By applying this shuffling noise, we want our model to learn to recognize and reorganize the information.
>
> (12) The generation has a hard limit, which is decided on the validation dataset. For instance, the maximum generation length for CNN/DM dataset is 175. Also, in beam search, if the generated token is <EOS>, i.e. the end of sentence, then the generation is terminated immediately for the current sequence.
>
> (13) Since TED is an abstractive model, this experiment is to show that TED has the ability to summarize using words/phrases not in the original article, which is typical in human-edited summaries. Explanations why TED has more novel grams could be TED has seen more data during the pretraining phase than PGNet (which is only trained using in-domain data). Also PGNet uses RNN while TED leverages transformer. The more powerful modeling ability of transformer can also help. Also the major evaluation metrics is the ROUGE, on which TED shows competitive performances.

---

### Official Review · AnonReviewer1 · 2019-10-25
**Official Blind Review #1**

**Rating:** 6

**Review:**

The authors propose to improve abstractive summarization models by using pretrained embeddings, theme modeling and denoising.

They propose a very interesting idea: to leverage the lead bias in news article to build supervized summarization task from 21.4 M of articles. Details are given how to produce this supervized data using simple heuristics.

The  model is  train with a denoising loss, by introducing 2 types of noise (tokens from other article and sequence shuffle). Theme modeling is also introduced as a classification problem  (same as BERT) :  the system must learn to classify pairs of sentences from the same article and pairs from different articles.

Experiments are conducted on 3 datasets. The proposed model outperforms the other unsupervized abstractive models and provides results closed to unsupervized extractive models, with a metrics which favors extractive models. Ablation study shows that pretraining yields most of the impact, whereas improvements due to theme modeling and denoising loss are marginal.

In the Article example :
"in the wold"  ?

Conclusion :
- dataset-agnostic : I don't see why since the approach take advantage of the lead bias.
- "outperforms previous systems by significant margins" : excessive.


**Experience Assessment:**

I do not know much about this area.

**Review Assessment: Checking Correctness Of Derivations And Theory:**

N/A

**Review Assessment: Checking Correctness Of Experiments:**

I assessed the sensibility of the experiments.

**Review Assessment: Thoroughness In Paper Reading:**

N/A

---

> ### Author Response · Authors · 2019-11-12
> **To reviewer #1**
>
> Thanks for your comments. Please find our responses below.
>
> (1) We have corrected the typo.
> (2) Sorry for the confusion. We have removed the term “dataset-agnostic”. We were trying to point out that the pretraining technique generates one single model that achieves consistently good performance across all 3 test datasets.
> (3) Thanks for pointing that out. We have changed it to “TED outperforms previous unsupervised abstractive baselines”.
>
> Should you have any questions, we are very happy to answer them.

---

### Decision · Program_Chairs · 2019-12-19

**Decision:**

Reject

**Comment:**

This paper proposes an abstractive text summarization model that takes advantage of lead bias for pretraining on unlabeled corpora and a combination of reconstruction and theme modeling loss for finetuning. Experiments on NYT, CNN/DM, and Gigaword datasets demonstrate the benefit of the proposed approach.

I think this is an interesting paper and the results are reasonably convincing. My only concern is regarding a parallel submission that contains a significant overlap in terms contributions, as originally pointed out by R2 (https://openreview.net/forum?id=ryxAY34YwB). All of us had an internal discussion regarding this submission and agree that if the lead bias is considered a contribution of another paper this paper is not strong enough.

Due to space constraint and the above concern, along with the issue that the two submissions contain a significant overlap in terms of authors as well, I recommend to reject this paper.